# Influence of Consumption of Two Peruvian Cocoa Populations on Mucosal and Systemic Immune Response in an Allergic Asthma Rat Model

**DOI:** 10.3390/nu14030410

**Published:** 2022-01-18

**Authors:** Marta Périz, Maria J. Rodríguez-Lagunas, Francisco J. Pérez-Cano, Ivan Best, Santiago Pastor-Soplin, Margarida Castell, Malén Massot-Cladera

**Affiliations:** 1Secció de Fisiologia, Departament de Bioquímica i Fisiologia, Facultat de Farmàcia i Ciències de l’Alimentació, Universitat de Barcelona (UB), 08028 Barcelona, Spain; marta.perizto@gmail.com (M.P.); mjrodriguez@ub.edu (M.J.R.-L.); franciscoperez@ub.edu (F.J.P.-C.); malen.massot@ub.edu (M.M.-C.); 2Institut de Recerca en Nutrició i Seguretat Alimentària (INSA-UB), UB, 08921 Santa Coloma de Gramenet, Spain; 3Programa Cacao, Ingeniería Agroforestal, Facultad de Ciencias Ambientales, Universidad Científica del Sur, Lima 15842, Peru; spastor@cientifica.edu.pe; 4Unidad de Investigación en Nutrición, Salud, Alimentos Funcionales y Nutracéuticos, Universidad San Ignacio de Loyola, Lima 15024, Peru; 5Centro de Investigación Biomédica en Red de Fisiopatología de la Obesidad y la Nutrición (CIBEROBN), Instituto de Salud Carlos III, 28029 Madrid, Spain

**Keywords:** asthma, bronchoalveolar lavage fluid, eosinophil, methylxanthines, polyphenols, *Theobroma cacao*

## Abstract

Different cocoa populations have demonstrated a protective role in a rat model of allergic asthma by attenuating the immunoglobulin (Ig) E synthesis and partially protecting against anaphylactic response. The aim of this study was to ascertain the effect of diets containing two native Peruvian cocoa populations (“Amazonas Peru” or APC, and “Criollo de Montaña” or CMC) and an ordinary cocoa (OC) on the bronchial compartment and the systemic and mucosal immune system in the same rat model of allergic asthma. Among other variables, cells and IgA content in the bronchoalveolar lavage fluid (BALF) and serum anti-allergen antibody response were analyzed. The three cocoa populations prevented the increase of the serum specific IgG1 (T helper 2 isotype). The three cocoa diets decreased asthma-induced granulocyte increase in the BALF, which was mainly due to the reduction in the proportion of eosinophils. Moreover, both the OC and CMC diets were able to prevent the leukocyte infiltration caused by asthma induction in both the trachea and nasal cavity and decreased the IgA in both fecal and BALF samples. Overall, these results highlight the potential of different cocoa populations in the prevention of allergic asthma.

## 1. Introduction

Cocoa, derived from the beans of the *Theobroma cacao* L. tree, has been consumed by indigenous civilizations in Mesoamerica since 460 B.C. [1,2]. The religious rituals and medical uses of cocoa beverages by the Olmecs were adopted by the Mayas and Mexicas, expanding the trees’ cultivation and their uses and customs [3]. Despite the domestication of the cocoa tree in the Mexican region and its extensive uses, it is native to the Amazon basin, likely in the north-eastern area of Peru [4] and southeast Ecuador, where cocoa use was evidenced by three kinds of archaeological studies, i.e., cocoa starch grains, absorbed theobromine residues and ancient DNA, dating from approximately 5300 years ago recovered from Santa Ana-La Florida [5]. Geographically separated cultivars adapted to the local conditions and human selection and distribution of cocoa beans over the years have driven the development of multiple geographic and genetic populations of *Theobroma cacao* L. [6]. According to Thomas et al. [4], the locations with the highest richness of species from the genus *Theobroma* are in the Upper Amazon regions of north-eastern Peru, which contain not only high levels of genetic diversity but also the highest number of different genetic clusters. The knowledge of the biologic properties of such cocoa populations could be useful to farmers and producers in order to increase the added value of cocoa grain and its derivatives, especially artisan chocolates, nutraceutical derivatives and cosmetic supplies. Additionally, knowing the cocoa distribution can be helpful in securing a valuable resource using strategies such as an in situ germplasm bank (in farm), ex situ collections or in the course of creating so-called protected areas.

The bioactive compounds of cocoa include fiber (both soluble fiber, such as pectic substances, and insoluble fiber, such as Klason lignin) [7], polyphenolic compounds (mainly flavonoids, such as epicatechin, catechin and procyanidins) and methylxanthines (such as theobromine, caffeine and theophylline) [8,9,10]. Cocoa paste from different Peruvian regions has shown different content of phenolics, flavonoids and methylxanthines, and this fact contributes to different biological properties both in vitro and in vivo [11]. By means of the antioxidant capacity or other properties of these compounds, they exert cardioprotective effects by reducing pro-inflammatory mediators such as leukotrienes (LT), interleukin (IL-) 1β and tumor necrosis factor (TNF) α, thus improving endothelial function, decreasing platelet aggregation and improving cardiovascular health [2,12,13,14,15,16]. In addition, cocoa and cocoa-derived foods have been suggested to possess the potential to counteract cognitive decline and sustain cognitive abilities in patients at risk [17].

Besides the influence on cardiovascular and nervous systems, we have demonstrated the anti-inflammatory and immunomodulatory properties of cocoa-enriched diets [10,18,19]. In this regard, a flavonoid-enriched cocoa extract was able to decrease, in vitro, the secretion of inflammatory mediators such as TNF-α, monocyte chemoattractant protein-1 and nitric oxide (NO) by macrophages [20]. In addition, the anti-inflammatory effect of cocoa has been evidenced in rat models of acute and chronic inflammation [21,22]. Concerning the acquired immune system, a 10% cocoa-enriched diet in rats was able to downregulate serum specific IgM, IgG1, IgG2a and IgG2c antibodies after an immunization [23], which suggests a regulatory effect of a cocoa diet in T helper (Th) 2-immune responses. Moreover, cocoa intake in rats resulted in an increase in the proportion of spleen B lymphocytes [24] and in that of natural killer (NK) cells and Tγδ lymphocytes in mesenteric lymph nodes [25,26], which may be responsible for the attenuating the synthesis of antibodies [27,28]. Moreover, a decrease in IL-4 release was found in activated lymph node cells and splenocytes from rats fed cocoa [23,24,25]. Taking together the fact that IL-4 promotes IgE synthesis [29], which is reduced in rats fed cocoa, and the attenuation of serum antibodies induced by this diet, cocoa feeding may be beneficial in reducing some allergies. In fact, in several rat models, it has been demonstrated that a diet containing 10% cocoa reduced the levels of specific antibodies related to Th2 immune response, including serum specific IgE [30,31]. In agreement with these findings, an observational study in young people revealed that the percentage of allergic people was lower in those consuming moderate or high amounts of cocoa than in people with low cocoa intake [32].

In the context of allergic diseases, asthma is a heterogeneous condition induced by sensitization to environmental allergens, characterized by respiratory symptoms with airflow limitation associated with airway inflammation and airway remodeling [33]. Allergic asthma results from sensitization to an allergen and a Th2-mediated response involving, among others, Th2 cells, basophils, eosinophils, mast cells and their major cytokines [34]. The activation and differentiation of allergen-specific Th2 cells induces the IgE synthesis against the allergen. This antibody binds to receptors on mast cells, which will be activated by subsequent exposures to the allergen, inducing mast cell degranulation, thus releasing histamine and proteases and producing the synthesis of prostanoids and LT [35]. Later, eosinophils will be recruited into the lungs, where a persistent airway inflammation occurs due to the subsequent exposures to the allergen [34]. Eosinophils contribute to the development of bronchial asthma via the release of mediators, such as LT, which will maintain the eosinophilic inflammation [36].

We have recently demonstrated that a native Peruvian cocoa population (CMC) exerted a protective effect on a rat model of allergic asthma due to an attenuating effect on IgE synthesis and the release of mast cell protease and a partial protection against anaphylactic response [11]. Nevertheless, it remained to be known what happened in both the mucosal and the systemic immune systems. In consequence, the aim of this study was to ascertain the effect of two diets containing two different native Peruvian cocoa populations on the bronchial immune system and the systemic compartment in the same rat model of allergic asthma.

## 2. Materials and Methods

### 2.1. Diets

Two native Peruvian cocoa pastes of different origins were used: “Amazonas Peru” cocoa paste (APC) from the Amazonas Region (latitude/longitude −5.737422, −78.431114), and “Criollo de Montaña” cocoa paste (CMC) from the Junín Region (−11.335774, −74.533181). In addition, CCN-51 ordinary cocoa paste (OC) obtained from the Cusco Region (−12.510664, −73.834577) was used as control. The different content of polyphenols, flavonoids, methylxanthines and the antioxidant capacity of these three cocoa populations have already been reported [11]. From these cocoa pastes, three diets were made by mixing 90% of powdered AIN-93M (Envigo, Huntingdon, UK) with 10% of cocoa paste previously pulverized. The mixture was pelletized, dried (40 °C for 36 h), vacuum-packed and stored at 4 °C until used. A standard diet based on the AIN-93M diet (Envigo) was used as reference. The composition of the diets is detailed in Table 1.

### 2.2. Animals and Allergic Asthma Induction

Female Brown Norway rats (4-week-old) were obtained from Envigo and were kept in the animal facilities at the Faculty of Pharmacy and Food Science (University of Barcelona (UB), Barcelona, Spain) in polycarbonate cages containing bedding of large fibrous particles under controlled conditions of temperature and humidity in a 12/12 h light/dark cycle. Procedures carried out in these animals were approved by the Ethical Committee for Animal Experimentation of the University of Barcelona (CEEA/UB ref. 414/16) and the Catalonia Government (DAAM 9351), following the EU-Directive 2010/63/EU for the protection of animals used for scientific purposes. 

The animals were randomized to achieve the same body weight mean into five experimental groups (9 animals/group). Throughout all the experimental procedure, two of the groups were fed with the reference diet and three groups received the three cocoa diets. Food and water were provided ad libitum. One week after the beginning of the dietetic intervention, allergic asthma was induced in one group fed the reference diet and the three groups fed cocoa diets. The procedure of asthma induction with ovalbumin (OVA) has already been reported [11,37]. Briefly, rats were intraperitoneally (i.p.) sensitized with 50 µg OVA (grade V, Sigma-Aldrich, Madrid, Spain), 2.5 mg aluminum hydroxide (alum; Imject^®^; Pierce, IL, USA) and 50 ng *Bordetella pertussis* toxin (Sigma-Aldrich). One week later, they received a booster (i.p.) with 50 µg OVA and 2.5 mg alum. After 3 weeks, all the animals were challenged with 300 µL of OVA solution (50 mg/mL) via intranasal (i.n.) instillation. Twenty-four hours after the challenge, the rats were anesthetized with ketamine (90 mg/kg) (Merial Laboratories S.A, Barcelona, Spain) and xylazine (10 mg/kg) (Bayer A.G, Leverkusen, Germany).

Blood samples were collected from the heart in ethylenediaminetetraacetic acid (EDTA)-anticoagulated tubes and in non-anticoagulated tubes. The EDTA blood sample was used to determine total and differential leukocyte counts using an automated hematology analyzer (Spincell 3, MonLab Laboratories, Barcelona, Spain). Non-anticoagulated blood was centrifuged, and the serum was kept at –20 °C until anti-OVA antibodies determination using the ELISA technique. Fecal samples were also collected and kept at −20 °C before obtaining the fecal homogenates by mixing fecal samples with phosphate buffered saline (PBS) (50 mg/mL), homogenizing them with a tissue homogenizer (Pellet Pestle Cordless Motor, Kimble, Meiningen, Germany) and finally centrifuging (300× *g*, 5 min, 4 °C). Fecal supernatants were used to quantify mucosal IgA.

### 2.3. Bronchoalveolar Lavage Fluid (BALF) Collection and Cellular Assessment

BALF collection was carried out as reported previously [37]. Briefly, 5 mL of chilled PBS was instilled through the trachea into the lungs and kept there for 30 s; then, BALF was retrieved. This was repeated three times with the same PBS and done again with an additional 5 mL PBS. The whole volume of the collected BALF (about 10 mL) was centrifuged at 538× *g* at 4 °C, and supernatants were distributed in aliquots and stored at –80 °C or –20 °C until cytokine and IgA quantification, respectively. Cell pellets were resuspended in 100 µL of PBS and used to establish the proportion of leukocytes using a hematology analyzer and by flow cytometry.

BALF suspension (20 µL) were used to establish leukocytes counts and the relative proportion of lymphocytes, monocytes and granulocytes using an automated hematology analyzer (MonLab Laboratories). In addition, the remaining BALF cells were also phenotyped via immunofluorescence staining and flow cytometry analysis to establish the proportions of B and T lymphocytes, dendritic cells, eosinophils and alveolar macrophages. The immunofluorescence staining procedure was carried out as in previous studies [38]. The antibodies used were specific to rat CD45, TCRαβ, CD45RA, CD11b and CD170 (BD Biosciences, Madrid, Spain) and were conjugated either to phycoerythrin, peridinin-chlorophyll a-protein, allophycocyanin or brilliant violet 421. Briefly, fresh BALF cells were incubated with fluorochrome-conjugated antibodies at saturating concentrations (20 min, 4 °C, in darkness). After washing the excess of antibodies, cells were fixed with 0.5% p-formaldehyde and stored at 4 °C protected from light until analysis (GalliosTM Cytometer, Beckman Coulter, Miami, FL, USA) in the Flow Cytometry Unit of the Scientific and Technological Centers of the University of Barcelona (CCiT-UB). Data were analyzed with Flowjo v10 software (Tree Star, Inc., Ashland, OR, USA). BALF CD45^+^ leukocytes were gated and then the proportion of T cells (TCRαβ^+^ CD45RA^−^), B cells (TCRαβ^−^ CD45RA^+^), dendritic cells (CD170^−^ CD11b^+^), eosinophils (CD170^+^ CD11b^+^) and alveolar macrophages (CD170^+^ CD11b^¯^) was assessed. 

### 2.4. BALF Cytokine Determination

The BALF concentration of IL-1α, IL-1β, IL-2, IL-4, IL-5, IL-6, IL-10, IL-12p70, IL-13, IL-17α, granulocyte colony-stimulating factor (G-CSF), granulocyte-macrophage colony-stimulating factor (GM-CSF), interferon gamma (IFN-γ), monocyte chemotactic protein (MCP)-1 and TNF-α was assessed using ProcartaPlex^®^ Multiplex Immunoassay following the manufacturer’s instructions (Affymetrix, eBioscience, San Diego, CA, USA). The concentration of each cytokine was obtained using a MAGPIX^®^ analyzer (Luminex Corporation, Austin, TX, USA) in the Flow Cytometry Unit of the CCiT-UB. The limits of detection were: 10 pg/mL for IL-1α, 13 pg/mL for IL-1β, 1.82 pg/mL for IL-2, 0.62 pg/mL for IL-4, 1.42 pg/mL for IL-5, 2.19 pg/mL for IL-6, 6.01 pg/mL for IL-10, 4.22 pg/mL for IL-12p70, 3 pg/mL for IL-13, 2.09 pg/mL for IL-17α, 4.91 pg/mL for G-CSF, 4.81 pg/mL for GM-CSF, 3.34 pg/mL for IFN-γ, 15 pg/mL for MCP-1 and 2.88 pg/mL for TNF-α.

### 2.5. BALF and Fecal IgA Content Quantification

BALF and fecal homogenate supernatants were used to assess the mucosal IgA concentration using a rat IgA ELISA Quantification Set (A110-102) following the manufacturer’s instructions (Bethyl Laboratories, Montgomery, TX, USA) as carried out in previous studies [39]. All samples were tested in duplicates. Absorbance was measured using a microplate photometer (Labsystems Multiskan, Helsinki, Finland) and data were interpolated using ASCENT version 2.6 software (Thermo Fisher Scientific, S.L.U, Barcelona, Spain) into standard curves. Results were expressed as fold change considering the mean value of the asthmatic group as 100%.

### 2.6. Serum Specific Anti-OVA Antibodies Quantification

The concentration of anti-OVA antibodies belonging to IgG1, IgG2a, IgG2b, IgG2c, and IgM in serum was quantified using an indirect ELISA as previously described [40]. All samples were tested in duplicates. For the quantification of specific IgG1 and IgG2a antibodies, serum was diluted 1/400,000; in the quantification of IgG2b and IgG2c, antibodies serum was diluted 1/1600. To quantify anti-OVA IgM, samples were diluted 1/800. Results were expressed as fold change considering the mean value of the asthmatic group as 100%.

### 2.7. Histopathological Examination

For the histological analysis, the upper part of the trachea and the nasal cavity were excised from anesthetized rats. The tissues were fixed in 4% buffered formaldehyde at room temperature for 24 h. Then, tissues were rinsed in PBS and dehydrated in graded ethanol (70%, 90% and 100%), and after permeation in xylene the samples were finally embedded in paraffin (Merck, Madrid, Spain). All samples were cut into 5 µm sections, stained with hematoxylin and eosin or periodic acid-Schiff (PAS) and photographed under a light microscope (Olympus BX41 and Olympus XC50 camera at 100×).

### 2.8. Statistical Analysis

The Statistical Package for the Social Sciences (SPSS v27.0, IBM, Chicago, IL, USA) was used for statistical analysis. Data were tested for homogeneity of variance and normality distribution using the Levene and Shapiro–Wilk tests, respectively. When data were homogeneous and had a normal behavior a conventional two-way ANOVA test followed by Bonferroni post hoc was performed. Otherwise, the nonparametric Kruskal–Wallis test was applied, followed by a post hoc analysis with the Mann–Whitney U test. Results were expressed as the mean ± SEM and significant differences were considered when *p* < 0.05. 

## 3. Results

### 3.1. Effect of Allergic Asthma and Cocoa Diets on Body Weight 

Before the intervention, the rat body weight ranged between 44 and 60 g, and at the end of the study the body weight had increased by about 95–130% (Figure 1). There was no significant difference in body weight gain due to either the asthma induction or nutritional interventions. Nevertheless, a tendency to a lower body weight gain was observed in all asthmatic animals compared to the healthy ones. 

### 3.2. Influence of Allergic Asthma and Cocoa Diets on Serum Specific Antibodies

The allergic asthma model induced the synthesis of anti-OVA antibodies belonging to IgM, IgG1, IgG2a, IgG2b and IgG2c isotypes (Figure 2) (*p* < 0.003, A group vs. REF group in all antibodies). The Th2-related antibodies (IgG1 and IgG2a isotypes) were the ones detected in the highest relative concentration because a higher dilution was needed for quantifying them (1/400,000 in both cases). 

The diets containing cocoa did not prevent the synthesis of either serum anti-OVA IgM, IgG2b and IgG2c antibodies. However, with regard to Th2-related antibodies, the three cocoa-enriched diets prevented the increase of anti-OVA IgG1 antibodies (*p* < 0.02, groups OC, APC and CMC vs. group A) by about 70–80%. Moreover, the APC cocoa diet prevented the increase in anti-OVA IgG2a by more than 60% (*p* = 0.019, group APC vs. group A), whereas both the OC and CMC diets showed a tendency to reduce its levels. 

### 3.3. Effects of Allergic Asthma and Cocoa Diets on IgA Concentration in Mucosal Compartments

The concentration of IgA at two mucosal compartments, i.e., the lung and the intestine, was determined by quantifying it in the BALF and fecal homogenates, respectively (Figure 3). The levels of mucosal IgA in both compartments did not differ between reference animals and asthmatic animals, however, the cocoa intake modified them. In particular, the OC and CMC diets were able to decrease the IgA levels in both mucosal compartments compared to REF and A groups (*p* < 0.02). 

### 3.4. Influence of Allergic Asthma and Cocoa Diets on Blood Leukocyte Counts

Total and differential blood leukocyte counts were established at the end of the study, i.e., 24 h after the i.n. challenge (Table 2). The values of total blood leukocyte counts did not differ between groups; however, when considering the differential leukocytes, asthma induction resulted in a reduction of blood lymphocyte proportion and an increase in granulocyte percentage. Diets containing cocoa did not prevent these changes. With regard to monocytes, there was no significant change due to asthma induction, but the monocyte percentage in animals fed OC and APC was lower than that found in healthy animals (Table 2).

### 3.5. Effects of Allergic Asthma and Cocoa Diets on BALF Leukocytes

Total leukocyte counts as well as lymphocyte, monocyte and granulocyte counts were quantified in BALF samples using an automated hematology analyzer and further by flow cytometry (Table 3 and Figure 4).

The counts of leukocytes recovered from BALF in asthmatic animals fed a standard diet were higher than those observed in REF animals (*p* = 0.047 group A vs. group REF). Such an increase was prevented by the OC diet but not by the APC and CMC diets (Table 3). The increase in leukocyte counts in asthmatic animals fed a standard diet as well as APC and CMC diets was due to higher counts of lymphocytes, monocytes and granulocytes because no changes in the proportion of such cells were observed in these groups (Table 3).

BALF lymphocyte subsets were established using flow cytometry (Figure 4a). The proportion of T cells was significantly reduced due to the asthma induction (*p* < 0.05 group A vs. group REF). This decrease was also observed in the APC-fed asthmatic animals (*p* < 0.02 group APC vs. group REF) but it was not in asthmatic animals fed other cocoa diets. The proportion of B cells was not modified by either allergic asthma induction or cocoa diets (Figure 4b).

The BALF proportion of dendritic cells, eosinophils and alveolar macrophages was also determined using flow cytometry (Figure 4b). Asthma induction did not modify the proportion of dendritic cells and alveolar macrophages, whereas that of eosinophils increased in asthmatic animals fed a standard diet (*p* < 0.05 group A vs. group REF). This increase was partially prevented by all three cocoa experimental diets (Figure 4b).

### 3.6. Effects of Allergic Asthma and Cocoa Diets on BALF Cytokines

Of the cytokines determined in BALF samples, only MCP-1, IL-10 and IL-13 achieved detectable levels. The BALF concentration of MCP-1 was not significantly modified by asthma induction or cocoa diets (Figure 5). Nevertheless, the BALF concentration of IL-10 increased in the asthmatic group (*p* < 0.05 vs. REF group) and also in OC- and APC-fed animals. Nevertheless, the CMC diet was able to prevent this increase (*p* < 0.05 CMC group vs. A, OC and APC groups). In contrast, the BALF concentration of IL-13 decreased in asthmatic animals fed the standard diet and also in those fed with the OC diet (*p* < 0.05 REF group vs. A and OC groups). This decrease was prevented by the CMC diet and partly by the APC diet.

### 3.7. Effects of Allergic Asthma and Cocoa Diets on Nasal Tissue and Trachea

The nasal cavity of asthma animals presented high leukocyte infiltrate, as it can be observed in hematoxylin and eosin stained sections (Figure 6a). The OC- and APC-fed animals showed a pattern more similar to REF animals. However, in the CMC group, even though there was some amelioration compared to the A animals, an increased presence of polymorphonuclear leukocytes (PMNs) was observed.

Trachea of REF animals displayed a pseudostratified ciliated epithelium, submucosal glands that add moisture to the air and aid in trapping contaminants, and hyaline cartilage (Figure 6b). The trachea of asthmatic animals presented leukocyte infiltrate with eosinophils and other PMNs. The OC- and CMC-fed animals showed a pattern more similar to REF animals. However, in the APC-fed ones, even showing some amelioration with respect to the A group, an increased presence of PMNs was observed. 

With regard to PAS staining, which stains mucus bright pink, asthmatic animals had more presence of mucus in the nasal cavity (Figure 6c) and in the trachea (Figure 6d) compared to REF animals. APC-fed animals seemed to revert this increase followed by the OC and CMC animals. 

## 4. Discussion

In a previous study we demonstrated the partial protective effect of native Peruvian cocoa populations on the anaphylactic response induced in a rat model of allergic asthma [11]. The aim of the current study was to go in depth into the systemic and mucosal immune responses of these animals.

Firstly, focusing on the systemic immune response, and in particular on the blood compartment, asthma induction produced changes in the proportions of blood leukocytes, decreasing the percentage of lymphocytes and increasing that of granulocytes, which included neutrophils, eosinophils and basophils. These alterations were not prevented by any cocoa diet. On the other hand, the asthma induction brought on the synthesis of specific antibodies belonging to IgM, IgG1, IgG2a, IgGb and IgG2c isotypes, which can be associated with either rat Th1 or Th2 immune responses [23]. Cocoa diets differentially affected the synthesis of these antibodies. While no significant changes were found in anti-OVA IgM and Th1-related IgG isotypes (i.e., IgG2b and IgG2c), the three cocoa diets attenuated anti-OVA IgG1 and IgG2a levels, associated with a Th2 immune response, although the effect on IgG2a only achieved statistical significance after APC diet intake. These results are in line with those obtained in the previous study in which we observed that serum anti-OVA IgE antibodies decreased in OC- and CMC-fed animals, whereas the APC diet did not significantly decrease IgE levels [11]. The distinct results obtained with the various cocoas could be attributed to a differential profile of bioactive compounds in the three cocoa populations, which differed in the amount of flavonoids and methylxanthines (CMC > APC > OC in both cases). Overall, the current results are in agreement with previous studies reporting the modulating effect of conventional cocoa on specific Th2-related antibodies in rat models of allergy [26,30,41]. With regard to the mechanisms involved in any cocoa intake regarding allergic response, they could include both the first phases of the acquired immune response, such as antigen presentation, and the cytokines secreted by effector T cells, as has been suggested [18]. 

Focusing on mucosal immunity, results regarding cell infiltration in the respiratory tissue and bronchoalveolar exudates show that asthma induction caused a higher presence of leukocytes, mainly granulocytes, as observed in nasal and tracheal sections and also in the BALF counts. In addition, flow cytometry analysis indicated a higher proportion of eosinophils and a lower proportion of T lymphocytes. These results agree with those reported by Thakur et al. [42] who found increased numbers of leukocytes, neutrophils, eosinophils, lymphocytes and monocytes in BALF 24 h after the challenge, in a similar model of rat asthma. Today, eosinophils are considered to play important roles in the development of asthma [36]. These cells can accumulate in respiratory airways after adhesion to vascular endothelial cells by means of α4 integrin/vascular cell adhesion molecule (VCAM)-1, which is upregulated by Th2 cytokines such as IL-4 and IL-13 [36]. In the lungs, eosinophils release a number of mediators, such as major basic protein (MBP), radical oxygen species (ROS), cytokines and leukotrienes [36]. Moreover, eosinophils promote airway edema, prompting both the recruitment and function of Th2 cells as well as mast cells. Therefore, the control of lung eosinophils could be a good strategy for controlling asthma. We found that cocoa diets decreased the granulocyte counts and particularly those of eosinophils in the BALF and also in airway tissues. The results regarding the decrease in eosinophil lung infiltration agree with those reported in rodent asthma models orally treated with natural extracts rich in flavonoids [43,44,45], or particular flavonoids [46,47]. All these results suggest that cocoa flavonoid content could be responsible for the protection against eosinophil infiltration. In addition, the effect on controlling eosinophils could be partially responsible for the protection of the anaphylactic response quantified by a decrease in body temperature and body weight, as well as by plasma mast cell protease II and BALF leukotrienes in these animals, as previously reported [11]. Furthermore, future experiments should analyze in depth the role of eosinophils in this asthma model and the cocoa effect by quantifying more specific eosinophils mediators.

When considering cytokine levels in the BALF, from the wide panel we applied, only the inflammatory MCP-1, the regulatory IL-10 and the Th2-related IL-13 were detected. In asthmatic animals fed standard diet, IL-10 increased while IL-13 decreased. The results of the cytokine contents do not agree with the higher levels of IL-4, IL-5 and IL-13 reported in other asthma models [46,48,49]. We did not detect IL-4 and IL-5, and there were surprisingly low concentrations of IL-13. The problem could be due to the use of the high volume of PBS instilled to obtain the BALF. We used a total of 10 mL per rat when others used 0.5–1 mL per mouse [46,49]. Further studies should consider the instillation of lower volumes of washing buffer to better detect cytokines in the BALF or even the application of more sensitive methods to better define the cytokine profile in healthy and asthmatic rats. In spite of the low levels obtained, it is worth highlighting the results obtained with the CMC diet, which was able to prevent the changes induced by asthma induction.

Finally, considering mucosal immunity, we also analyzed the total IgA content in two mucosal compartments: the BALF and the intestine (fecal samples). It has been suggested that IgA is a protective antibody because there is an inverse relationship between mucosal IgA levels and the incidence of allergic airway disease [50]. In our study, no changes in BALF IgA content in asthmatic animals were found. These results agree with data reported in a mouse model of allergic asthma also induced with OVA [51], however, they do not concur with another study in mice with allergic asthma induced using a fungal biopesticide showing an IgA increase [52]. Although the BALF IgA content was not influenced by allergy asthma induction in our model, the OC and CMC diets—but not the APC diet—decreased it. The results of cocoa diets on the BALF IgA levels match those of fecal IgA content and agree with previous studies using diets with 10% conventional cocoa, which induced lower IgA values in salivary glands [39] and in the intestine [18,25,26,39,53,54]. The reason why the APC diet did not affect mucosal IgA remains unknown, and it cannot be attributed to the total amount of flavonoids and methylxanthines because it contains levels ranging between those of the CMC and OC populations [11]. Nevertheless, the difference between the effects of cocoa could derive from a particular flavonoid profile, which depends on the origin of the cocoa. The content of flavonoids and methylxanthines is higher in the CMC population, from the Junín region (Satipo city) in Peru, with a hot and humid climate, than in the APC population, coming from the Amazonas region [11]. Previous studies have appointed epicatechin as being partially responsible for the effect of cocoa diet on fecal IgA levels due to an inverse correlation between fecal IgA concentration and the quantity of epicatechin derivatives in urine [55] without showing a dose-response relationship [56,57]. However, more recently, cocoa theobromine has been considered the main compound responsible for decreasing intestinal IgA [55,58]. Therefore, differences between APC and CMC populations could be due to the content of both theobromine and flavonoids. However, as the OC diet caused a similar effect to that of CMC and its content in these compounds was lower [11], this suggests that other factors must also have an influence. In fact, these results together with other differential values obtained for the three cocoa populations prompt us to establish, in further studies, the exact chemical profile of each cocoa paste and to correlate them with the preventive effect found here.

## 5. Conclusions

The results regarding antiallergen IgG antibodies and the changes in the cellularity observed in the bronchoalveolar lavage fluid and in the airway pathway tissues highlight the potential of different populations of Peruvian cocoa in the prevention of allergic asthma. This could be important in countries with a high prevalence of asthma, such as Peru, where localities with a high genetic diversity of the *Theobroma cacao* L. trees are found.

## Figures and Tables

**Figure 1 nutrients-14-00410-f001:**
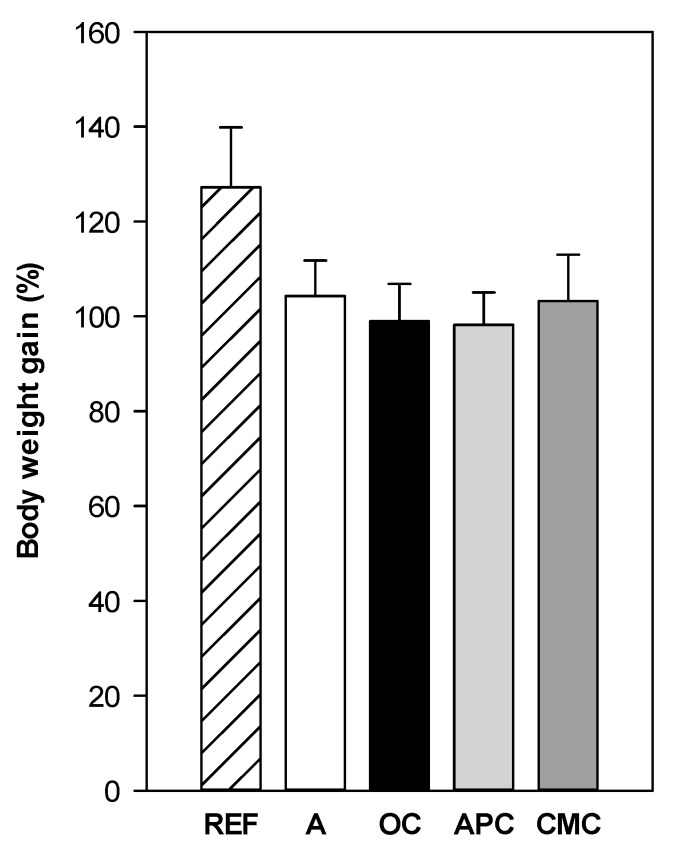
Body weight gain compared to that at the beginning of the nutritional intervention. REF: healthy animals fed standard diet; A: asthmatic animals fed standard diet; OC: asthmatic animals fed ordinary cocoa-enriched diet; APC: asthmatic animals fed “Amazonas Peru” cocoa-enriched diet; CMC: asthmatic animals fed “Criollo de Montaña” cocoa-enriched diet. Results are shown as mean ± standard error (N = 9).

**Figure 2 nutrients-14-00410-f002:**
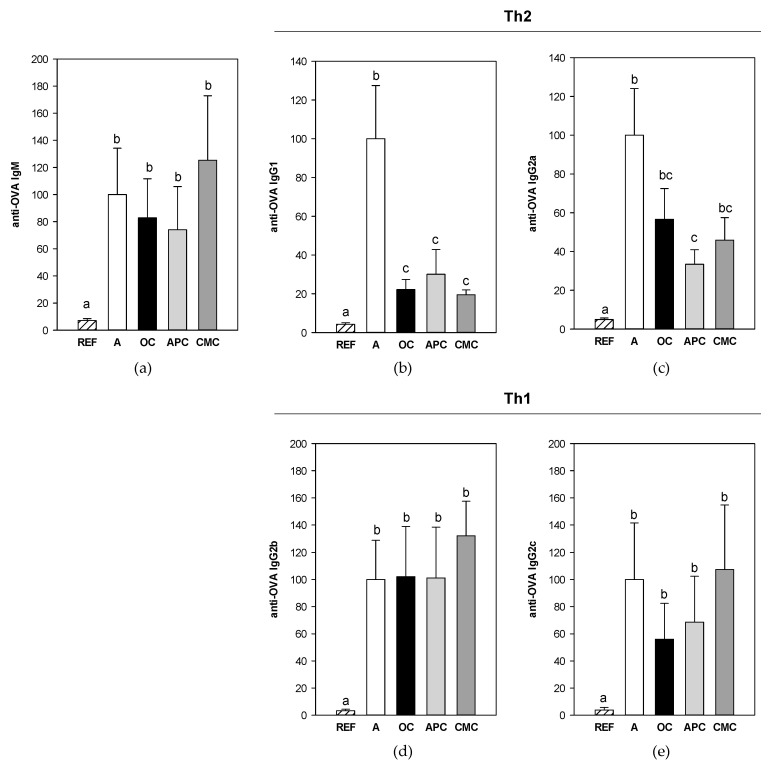
Relative concentration of anti-OVA IgM (**a**), Th2-related (IgG1 and IgG2a) (**b**,**c**) and Th1-related (IgG2b and IgG2c) (**d**,**e**) antibodies in serum obtained 24 h after the intranasal challenge. All experimental group results were referred to group A levels, which were considered as 100%. REF: healthy animals fed standard diet; A: asthmatic animals fed standard diet; OC: asthmatic animals fed with ordinary cocoa-enriched diet; APC: asthmatic animals fed with “Amazonas Peru” cocoa-enriched diet; CMC: asthmatic animals fed with “Criollo de Montaña” cocoa-enriched diet. Results are shown as mean ± standard error (N = 9). Values not sharing letters denote significant differences between groups (*p* < 0.05), while values sharing the same letter do not differ.

**Figure 3 nutrients-14-00410-f003:**
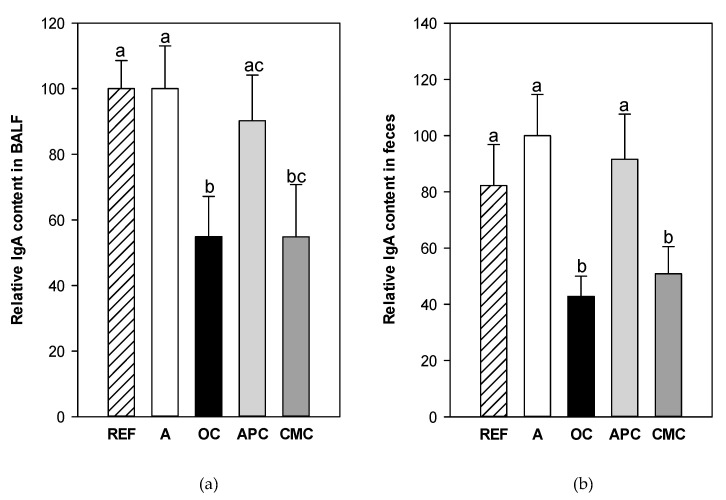
Relative IgA content in the bronchoalveolar lavage fluid (**a**) and feces (**b**) in samples obtained 24 h after the intranasal challenge. All experimental group results were referred to group A levels which were considered as 100%. REF: healthy animals fed standard diet; A: asthmatic animals fed standard diet; OC: asthmatic animals fed with ordinary cocoa-enriched diet; APC: asthmatic animals fed with “Amazonas Peru” cocoa-enriched diet; CMC: asthmatic animals fed with “Criollo de Montaña” cocoa-enriched diet. Results are shown as mean ± standard error (N = 9). Values not sharing letters denote significant differences between groups (*p* < 0.05), while values sharing the same letter did not differ.

**Figure 4 nutrients-14-00410-f004:**
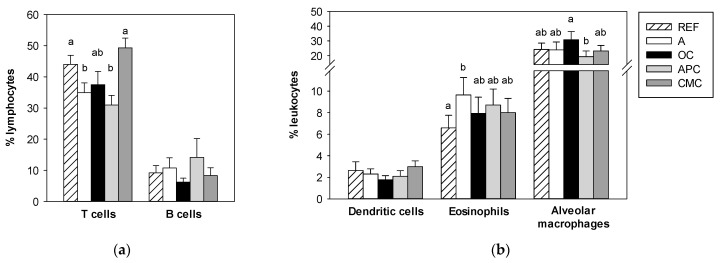
Proportion of lymphocyte subsets (**a**) and dendritic cells, eosinophils and alveolar macrophages (**b**) in bronchoalveolar lavage fluid (BALF) samples obtained 24 h after the intranasal challenge. REF: healthy animals fed standard diet; A: asthmatic animals fed standard diet; OC: asthmatic animals fed with ordinary cocoa-enriched diet; APC: asthmatic animals fed with “Amazonas Peru” cocoa-enriched diet; CMC: asthmatic animals fed with “Criollo de Montaña” cocoa-enriched diet. Results are shown as mean ± standard error (N = 9). Values not sharing letters denote significant differences between groups (*p* < 0.05), while values sharing the same letter did not differ.

**Figure 5 nutrients-14-00410-f005:**
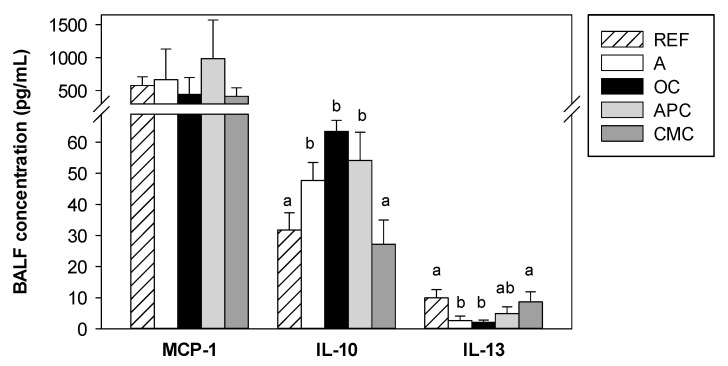
Cytokine concentration in bronchoalveolar lavage fluid (BALF) samples obtained 24 h after the intranasal challenge. REF: healthy animals fed standard diet; A: asthmatic animals fed standard diet; OC: asthmatic animals fed with ordinary cocoa-enriched diet; APC: asthmatic animals fed with “Amazonas Peru” cocoa-enriched diet; CMC: asthmatic animals fed with “Criollo de Montaña” cocoa-enriched diet. Results are shown as mean ± standard error (N = 9). Values not sharing letters denote significant differences between groups (*p* < 0.05), while values sharing the same letter did not differ.

**Figure 6 nutrients-14-00410-f006:**
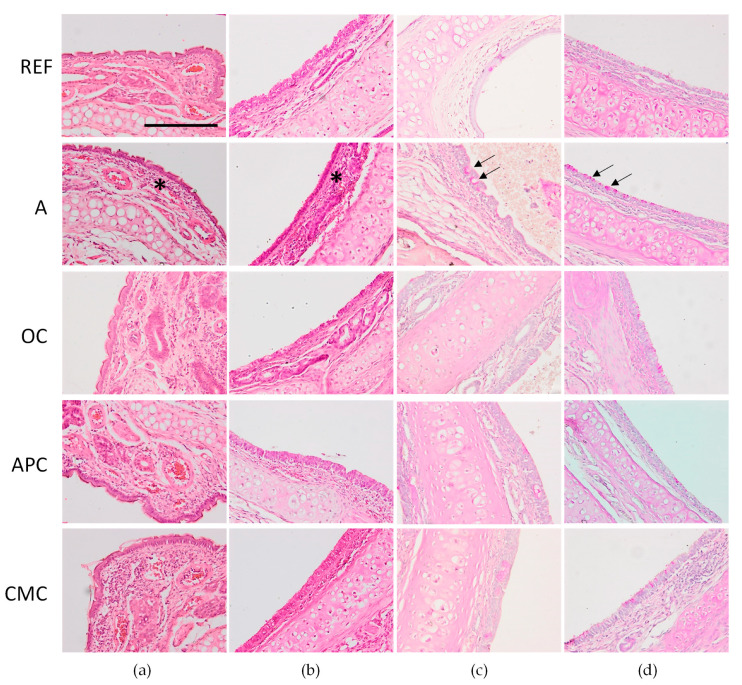
Hematoxylin and eosin staining (**a**,**b**) and PAS staining (**c**,**d**) of nasal tissue (**a**,**c**) and trachea (**b**,**d**) obtained 24 h after the intranasal challenge. REF: healthy animals fed standard diet; A: asthmatic animals fed standard diet; OC: asthmatic animals fed with ordinary cocoa-enriched diet; APC: asthmatic animals fed with “Amazonas Peru” cocoa-enriched diet; CMC: asthmatic animals fed with “Criollo de Montaña” cocoa-enriched diet. Scale bar = 100 µm. Asterisk indicates leukocyte infiltrate and arrow indicates presence of mucus.

**Table 1 nutrients-14-00410-t001:** Composition of the experimental diets.

	APC Diet	CMC Diet	OC Diet	REF Diet
Casein (g/kg)	126	126	126	140
L-Cystine (g/kg)	1.62	1.62	1.62	1.8
Corn Starch (g/kg)	419.12	419.12	419.12	466
Maltodextrin (g/kg)	139.5	139.5	139.5	155
Sucrose (g/kg)	90	90	90	100
Soybean oil (g/kg)	36	36	36	40
Cellulose (g/kg)	45	45	45	50
Mineral mix (g/kg)	31.5	31.5	31.5	35.0
Vitamin mix (g/kg)	9	9	9	10
Choline bitartrate (g/kg)	2.25	2.25	2.25	2.50
TBHQ (antioxidant) (g/kg)	0.007	0.007	0.007	0.008
Cocoa paste (g/kg):	100	100	100	-
Total phenolics (g gallic acid equivalents/kg)	2.87	3.04	2.41	-
Total flavonoids (g gallic acid equivalents/kg)	5.04	5.66	3.48	-
Total methylxanthines (g/kg)	0.841	0.966	0.799	-

APC diet: “Amazonas Peru” cocoa-enriched diet; CMC diet: “Criollo de Montaña” cocoa-enriched diet; OC diet: ordinary cocoa-enriched diet; REF diet: AIN-93M diet. TBHQ: ter-butyl hydroquinone.

**Table 2 nutrients-14-00410-t002:** Total and differential leukocyte counts in blood samples obtained 24 h after the intranasal challenge.

	REF Group	A Group	OC Group	APC Group	CMC Group
Total leukocyte counts (×10^9^/L)	11.22 ± 0.86	10.23 ± 1.69	10.76 ± 1.40	9.92 ± 1.08	10.08 ± 1.38
Lymphocyte (%)	70.79 ± 2.08 ^a^	64.03 ± 1.56 ^b^	63.07 ± 8.70 ^b^	64.26 ± 7.72 ^b^	59.03 ± 3.93 ^b^
Monocyte (%)	6.36 ± 0.15 ^a^	6.24 ± 0.21 ^ab^	5.96 ± 0.08 ^b^	5.84 ± 0.70 ^b^	5.91 ± 0.23 ^ab^
Granulocyte (%)	22.86 ± 2.07 ^a^	29.72 ± 1.72 ^b^	31.01 ± 4.68 ^b^	29.90 ± 3.85 ^b^	35.06 ± 3.98 ^b^

REF: healthy animals fed standard diet; A: asthmatic animals fed standard diet; OC: asthmatic animals fed with ordinary cocoa-enriched diet; APC: asthmatic animals fed with “Amazonas Peru” cocoa-enriched diet; CMC: asthmatic animals fed with “Criollo de Montaña” cocoa-enriched diet. Results are shown as mean ± standard error (N = 9). Values not sharing letters denote significant differences between groups (*p* < 0.05), while values sharing the same letter did not differ.

**Table 3 nutrients-14-00410-t003:** Total and differential leukocyte counts in bronchoalveolar samples obtained 24 h after the intranasal challenge.

	REF Group	A Group	OC Group	APC Group	CMC Group
Total leukocyte counts (×10^8^/L)	1.31 ± 0.10 ^a^	2.07 ± 0.31 ^b^	1.60 ± 0.15 ^ab^	2.47 ± 0.42 ^b^	2.42 ± 0.36 ^b^
Lymphocyte (%)	32.65 ± 2.43	32.43 ± 2.98	35.06 ± 3.85	46.35 ± 6.47	42.11 ± 4.58
Monocyte (%)	14.63 ± 0.77	12.39 ± 0.58	12.81 ± 0.56	12.32 ± 0.93	12.75 ± 1.43
Granulocyte (%)	52.72 ± 2.85	55.17 ± 2.88	52.13 ±3.97	41.33 ± 6.60	45.13 ± 4.77

REF: healthy animals fed standard diet; A: asthmatic animals fed standard diet; OC: asthmatic animals fed with ordinary cocoa-enriched diet; APC: asthmatic animals fed with “Amazonas Peru” cocoa-enriched diet; CMC: asthmatic animals fed with “Criollo de Montaña” cocoa-enriched diet. Results are shown as mean ± standard error (N = 9). Values not sharing letters denote significant differences between groups (*p* < 0.05), while values sharing the same letter did not differ.

## Data Availability

Not applicable.

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
