# Peer review of "Influence of Consumption of Two Peruvian Cocoa Populations on Mucosal and Systemic Immune Response in an Allergic Asthma Rat Model"

_nutrients, 2022, doi:10.3390/nu14030410_

Round 1

Reviewer 1 Report

Overall a fine study and paper. I wonder about the true significance of the findings, both in terms of explaining your previous results as regards a reduced potential for anaphylaxis, as well as applicability to human allergic disease. 

As you point out in your discussion I was also questioning the meaning of the reduced levels of IgA in the cocoa fed animals, given the apparent protective effect/inverse association of IgA and allergen sensitisation and airway hyper-responsiveness in other studies. The IgG results are more consistent with your argument and what would be expected given the study design.

It is also somewhat surprising that you were able to detect IL10 and IL13 (even in the reference group!) but not other cytokines which would typically be produced in greater quantities in this setting. The dilutional explanation in collection of BALF is noted, however, I am hesitant to draw too many conclusions from these cytokines and MCP-1 alone without being able to view them in context of the overall inflammatory profile. One does need to wonder why the non-asthmatic group is making more Th2 cytokine (IL13).

The opening paragraph of the introduction is focussed largely on the history, geographical distribution, production and commercial relevance of cocoa, without any mention of allergic disease. Consider rewriting the opening to reflect the purpose of the paper.

Author Response

Reviewer 1

Overall a fine study and paper. I wonder about the true significance of the findings, both in terms of explaining your previous results as regards a reduced potential for anaphylaxis, as well as applicability to human allergic disease. 

As you point out in your discussion, I was also questioning the meaning of the reduced levels of IgA in the cocoa fed animals, given the apparent protective effect/inverse association of IgA and allergen sensitisation and airway hyper-responsiveness in other studies. The IgG results are more consistent with your argument and what would be expected given the study design.

Answer:

It is true that the protective effects of IgA against allergic airway disease have been suggested in some studies (reference 50). However, in our experimental model, the asthma induction did not result in a reduction of its levels in those animals fed with standard diet which is in line with those results reported by reference 51. For this reason, we discussed about the disagreement with the published studies.

On the other hand, a significant decrease was observed in animals fed with two cocoa diets, which agrees with the IgA reduction induced by different cocoa sources in both intestinal and extraintestinal compartments (references 18,25,26,39,53,54). Furthermore, we previously observed from cocoa’s bioactive compounds (polyphenols, theobromine, fiber), the theobromine seems to be the main responsible for decreasing IgA content. As in the current article, there was a cocoa diet that did not have the same behavior, we discussed about it, but we could not conclude the reason of such disagreement.

In agreement with the reviewer, the levels of specific IgG in our model of allergy and the preventive role of cocoa in such increase are more relevant.

It is also somewhat surprising that you were able to detect IL10 and IL13 (even in the reference group!) but not other cytokines which would typically be produced in greater quantities in this setting. The dilutional explanation in collection of BALF is noted, however, I am hesitant to draw too many conclusions from these cytokines and MCP-1 alone without being able to view them in context of the overall inflammatory profile. One does need to wonder why the non-asthmatic group is making more Th2 cytokine (IL13).

Answer:

We agree with the reviewer that results about BALF cytokines are surprising and conflicting with we waited, and they do not allow to draw too many conclusions. However, we have added an alternative to better define the cytokine profile in further studies (lines 432-433).

The opening paragraph of the introduction is focused largely on the history, geographical distribution, production and commercial relevance of cocoa, without any mention of allergic disease. Consider rewriting the opening to reflect the purpose of the paper.

Answer:

We believe that starting the introduction with cocoa is better because the journal is Nutrients and because the paper is sent to the “Diet and Nutrition in Asthma and Allergic Disorders” Special Issue. Therefore, we consider that the introduction should start focusing on the bioactive compounds used in the nutritional intervention and we would prefer keeping the Introduction in its current order. However, if the reviewer still think that it is important to be published, we will modify it.

References

  1. Camps-Bossacoma, M.; Massot-Cladera, M.; Abril-Gil, M.; Franch, A.; Pérez-Cano, F.J.; Castell, M. Cocoa Diet and Antibody Immune Response in Preclinical Studies. Front. Nutr. 2017, 4, 1–14, doi:10.3389/fnut.2017.00028.
  2. Ramiro-Puig, E.; Pérez-Cano, F.J.; Ramos-Romero, S.; Pérez-Berezo, T.; Castellote, C.; Permanyer, J.; Franch, À.; Izquierdo-Pulido, M.; Castell, M. Intestinal immune system of young rats influenced by cocoa-enriched diet. J. Nutr. Biochem. 2008, 19, 555–565, doi:10.1016/j.jnutbio.2007.07.002.
  3. Camps-Bossacoma, M.; Abril-Gil, M.; Saldaña-Ruiz, S.; Franch, À.; Pérez-Cano, F.J.; Castell, M. Cocoa diet prevents antibody synthesis and modifies lymph node composition and functionality in a rat oral sensitization model. Nutrients 2016, 8, 1–17, doi:10.3390/nu8040242.
  4. Massot-Cladera, M.; Franch, À.; Pérez-Cano, F.J.; Castell, M. Cocoa and cocoa fibre differentially modulate IgA and IgM production at mucosal sites. Br. J. Nutr. 2016, 115, 1539–1546, doi:10.1017/S000711451600074X.
  5. 50. Gloudemans, A.K.; Lambrecht, B.N.; Smits, H.H. Potential of immunoglobulin A to prevent allergic asthma. Clin. Dev. Immunol. 2013, 2013, 542091.
  6. Kumar, R.K.; Temelkovski, J.; McNeil, H.P.; Hunter, N. Airway inflammation in a murine model of chronic asthma: Evidence for a local humoral immune response. Clin. Exp. Allergy 2000, 30, 1486–1492, doi:10.1046/j.1365-2222.2000.00911.x.
  7. Pérez-Berezo, T.; Franch, A.; Ramos-Romero, S.; Castellote, C.; Pérez-Cano, F.J.; Castell, M. Cocoa-enriched diets modulate intestinal and systemic humoral immune response in young adult rats. Mol. Nutr. Food Res. 2011, 55, doi:10.1002/mnfr.201000588.
  8. Pérez-Berezo, T.; Franch, A.; Castellote, C.; Castell, M.; Pérez-Cano, F.J. Mechanisms involved in down-regulation of intestinal IgA in rats by high cocoa intake. J. Nutr. Biochem. 2012, 23, 838–844, doi:10.1016/j.jnutbio.2011.04.008.

Reviewer 2 Report

Reviewer's revision of the manuscript. The authors have extensive experience in the discussed subject and in research methodology. Although the observations were not always consistent with the assumptions of the model - the methodological description of the obtained results, and in particular the discussion of these results and possible causes, were very interesting.

Below, I’ve presented some minor editorial comments, which, in the opinion of the reviewer, will facilitate the reading of the text of the manuscript for the reader.

Introduction

  1. Line: 54: In the opinion of the reviewer, the structure of this sentence should be changed a bit: “Additionally, to know the cocoa distribution could help conserve them by different strategies such as in situ germplasm bank (in farm), ex situ collections or protected areas”. proponujÄ™ na przykÅ‚ad na nastÄ™pujÄ…cy: “Additionally, to know the cocoa distribution can be helpful in securing a valuable resource using strategies such as in situ germplasm bank (in farm), ex situ collections or in the course of creating so-called protected areas”. Mniej wiÄ™cej w ten sposób. By this I mean there is a mental shortcut in the original sentence when it comes to "conserve" and "strategies". Perhaps it would be better to elaborate a little here on how? Please consider it.

Cz jest potrzebne tak szerokie nawiÄ…zanie do roli eozynofili w etiologii astmy? czy badano te biaÅ‚ka EPO, ECP skoro o nich mowa we wstepie? 

  1. Materials and Methods

2.2 Animals and allergic asthma induction

  1. Line: 137-149: in the description of the methodology it’s not explicitly stated that the animals were fed the mentioned investigated diets throughout the induction of the inflammatory process and the asthma-like symptoms. It has been only mentioned that: "One week after the beginning of the dietetic intervention, allergic asthma was induced in one group fed reference diet and the three groups fed cocoa diets" - which, of course, can be guessed as I mentioned. But maybe please consider whether or not it is worth emphasizing it in one sentence - just for the clarity of reading such information. 

2.3 Bronchoalveolar lavage fluid (BALF) collection and cellular assessment

  1. Line: 164-167: It is worth clarifying here, emphasizing why, as I understand it, two fractions of supernatants were collected (frozen, successively at -80 and -20C)? that is, which supernatant from which BALF fraction was derived.
  2. Line: 168-169: In the automatic determination procedure with the usage of hematology analyzer (MonLab Laboratories), I suggest adding at least the information what volume was used or the number of cells was intended for analysis? The description in these sentences seems a bit too laconic. Please consider it.

2.4 BALF cytokine determination

  1. What is the result of informing the results for the obtained concentrations for individual targets in the form without information about the error limit (SD or SEM)?

2.5 BALF and fecal IgA content quantification

  1. Line: 203: I propose to change the word 'are' to the past mode = 'were'.
  2. Why has the IgE quantification not been planned (especially since this globulin is mentioned in the Introduction chapter)? I kindly ask for a very short comment.

2.6 Serum specific anti-OVA antibodies quantification 

  1. Line: 210: Similarly as above, I’d propose to change the word 'are' to the past mode = 'were'.
  2. In the description of all analytical methodologies - quantification of individual proteins - please make sure whether the information is included whether the analyzes were performed in replicates and in how many? (which can be guessed from the tabular results with the description “±”.).

2.8 Statistical analysis

  1. Line: 226: In this section, I likewise propose to change the indicative mode in the word "are" to the past one: 'were'.

  1. Results

3.2 Influence of allergic asthma and cocoa diets on serum specific antibodies 

  1. Line: 255-261: In the opinion of the reviewer, the description of the degree of changes and significance of the results obtained in Figure 2 is not fully understandable, clear in reading for the reader - and thus gives the impression that it may not fully reflect the description of the results obtained in section 3.2. and vice versa. Yes, there is information that “Values ​​not sharing letters denote significant differences between groups (p <0.05), while values ​​sharing the same letter do not differ”. On the other hand, both the description in the legend of the figure and the marking of the bars do not reflect the degree of significance changes between the groups, both in relation to the negative control group (healthy animals), to the control asthmatic animals (not receiving any diet), and between individual groups of asthmatic rats under the influence of studied diets. Please consider whether under the graph it would be better to adopt such a system of description and notations on the graphs that would reflect the above-mentioned relationships (i.e. type <0.05; <0.005; <0.001 or something like that). 

For example, looking at the graph in Fig. 2 a - e - you can see clear differences in the bars between the CMC and control A groups (reflecting the pathological state model), not to mention that obviously relative to the REF group). The same seems to be true for APC vs A (a) or OC vs A (e). The question is whether the observed changes are statistically significant in relation to each other? What, in the eyes of the reader, does not seem fully legible from the chart, yet, is mentioned in the text.

However, this is just a suggestion to think about. The same considerations apply to the description of the results and to the legend below the figures in later sections of the manuscript.

  1. Discussion
  2. The authors observed an interesting relationship regarding the scale of cell infiltration in the respiratory tissues and bronchoalveolar exudates. A technical question arises at this point, have the authors considered analyzing significant, neutrophil-selective (and more of that cell fraction for which the most interesting relationships were observed) potential markers (i.e. mentioned in the manuscript: EPO, ECP)? If not, are they considering addressing this topic in future studies ?. These are, of course, questions to consider and do not require a separate entry in the text of the discussion.
  3. Perhaps it is worth considering including information in the discussion on the fact that the CMC diet had the most significant impact on the observed concentrations of the investigated (and detected) interleukins, which would rather be its advantage in this case.
  4. As a comment, I would like to ask if the authors are considering carrying out similar studies taking into account the exact photochemical profile of the individual components of the cocoa paste. What in the context of the discussion, as the authors rightly emphasized at the end of the chapter, in particular with regard to flavonoids (and other potential components), seems to be intriguing? Wouldn't you consider including this information somewhere at the end of this chapter?
  5. Isn't it also worth considering studies with a controlled dosing of the tested cocoa pastes by dosing them with an intragastric tube to animals? In this way, the researcher would have control over the application of the planned dose of each paste form?

Questions 3 and 4 are only comments for authors to consider.

Author Response

Reviewer 2

The authors have extensive experience in the discussed subject and in research methodology. Although the observations were not always consistent with the assumptions of the model - the methodological description of the obtained results, and in particular the discussion of these results and possible causes, were very interesting.

Below, I’ve presented some minor editorial comments, which, in the opinion of the reviewer, will facilitate the reading of the text of the manuscript for the reader.

Introduction

  1. Line: 54: In the opinion of the reviewer, the structure of this sentence should be changed a bit: “Additionally, to know the cocoa distribution could help conserve them by different strategies such as in situ germplasm bank (in farm), ex situ collections or protected areas”. proponujÄ™ na przykÅ‚ad na nastÄ™pujÄ…cy: “Additionally, to know the cocoa distribution can be helpful in securing a valuable resource using strategies such as in situ germplasm bank (in farm), ex situ collections or in the course of creating so-called protected areas”. Mniej wiÄ™cej w ten sposób. By this I mean there is a mental shortcut in the original sentence when it comes to "conserve" and "strategies". Perhaps it would be better to elaborate a little here on how? Please consider it.

Answer:

Following the reviewer’s suggestion, this sentence have been modified. Now it reads as follows in the new version of the manuscript (lines 54-57): “Additionally, to know the cocoa distribution can be helpful in securing a valuable resource using strategies such as in situ germplasm bank (in farm), ex situ collections or in the course of creating so-called protected areas”.

Cz jest potrzebne tak szerokie nawiÄ…zanie do roli eozynofili w etiologii astmy? czy badano te biaÅ‚ka EPO, ECP skoro o nich mowa we wstepie? 

Using Google translator we read: Why is such a broad reference to the role of eosinophils in the etiology of asthma needed? Have these EPO and ECP proteins been studied, as we are talking about them in the introduction?

Answer:

We think that introducing the role of eosinophils played in asthma in the Introduction will help understanding the results and the discussion. In fact, we previously quantified leukotrienes in the same model of allergy. Nevertheless, as we did not study eosinophils mediators in the current paper, we have summarized this sentence in the current version of the manuscript (lines 102-104).

  1. Materials and Methods

2.2 Animals and allergic asthma induction

  1. Line: 137-149: in the description of the methodology it’s not explicitly stated that the animals were fed the mentioned investigated diets throughout the induction of the inflammatory process and the asthma-like symptoms. It has been only mentioned that: "One week after the beginning of the dietetic intervention, allergic asthma was induced in one group fed reference diet and the three groups fed cocoa diets" - which, of course, can be guessed as I mentioned. But maybe please consider whether or not it is worth emphasizing it in one sentence - just for the clarity of reading such information. 

Answer:

To make it clearer and following the reviewer’s suggestion, the duration of the nutritional intervention has been clarified. Now it reads as follows in the new version of the manuscript (line 139): “Throughout all the experimental procedure, two of the groups were fed with reference diet and three groups received the three cocoa diets”.

2.3 Bronchoalveolar lavage fluid (BALF) collection and cellular assessment

  1. Line: 164-167: It is worth clarifying here, emphasizing why, as I understand it, two fractions of supernatants were collected (frozen, successively at -80 and -20C)? that is, which supernatant from which BALF fraction was derived.

Answer:

The BALF was obtained by instilling twice 5 mL of chilled PBS through the trachea into the lungs to a final volume of 10 mL PBS. Once the 10 mL of BALF were obtained, the BALF was centrifuged, and the supernatant was collected and stored either at -80 ºC (for cytokine quantification) or -20 ºC (for IgA quantification). There were not two different fractions. To clarify this in the paper, we have rewritten this sentence (lines 166-169).

  1. Line: 168-169: In the automatic determination procedure with the usage of hematology analyzer (MonLab Laboratories), I suggest adding at least the information what volume was used or the number of cells was intended for analysis? The description in these sentences seems a bit too laconic. Please consider it.

Answer:

Following the reviewer’s suggestion, we have added new information about the volume of the BALF used in these techniques (lines 171-173).

2.4 BALF cytokine determination

  1. What is the result of informing the results for the obtained concentrations for individual targets in the form without information about the error limit (SD or SEM)?

Answer:

The values stated in this section of the manuscript refer to the limits of detection for each cytokine that depend on the batch of the kit used (ProcartaPlex Multiplex Immunoassay). They do not refer to the results obtained after the determination in our samples. In fact, as it can be read in lines 333-334, of the cytokines determined in BALF samples, only MCP-1, IL-10 and IL-13 achieved detectable levels.

2.5 BALF and fecal IgA content quantification

  1. Line: 203: I propose to change the word 'are' to the past mode = 'were'.

Answer:

Following the reviewer’s suggestion, the verb form has been changed.

  1. Why has the IgE quantification not been planned (especially since this globulin is mentioned in the Introduction chapter)? I kindly ask for a very short comment.

Answer:

As we stated at the end of the Introduction section, we have already demonstrated in the same rat model that a Peruvian cocoa population was able to attenuate the synthesis of specific IgE. We have not repeated it in the current paper. This is already included in both Introduction (lines 106-108) and Discussion (lines 387-389) sections.

2.6 Serum specific anti-OVA antibodies quantification 

  1. Line: 210: Similarly as above, I’d propose to change the word 'are' to the past mode = 'were'.

Answer:

Following the reviewer’s suggestion, the verb form has been changed.

  1. In the description of all analytical methodologies - quantification of individual proteins - please make sure whether the information is included whether the analyzes were performed in replicates and in how many? (which can be guessed from the tabular results with the description “±”.).

Answer:

Following this suggestion, we have added that ELISA technique for all samples in the anti-OVA antibodies quantification was carried out in duplicates (line 214). The same was added for IgA quantification (line 206). All techniques were performed in the 9 animals of each group.

2.8 Statistical analysis

  1. Line: 226: In this section, I likewise propose to change the indicative mode in the word "are" to the past one: 'were'.

Answer:

Following the reviewer’s suggestion, the verb form has been changed.

  1. Results

3.2 Influence of allergic asthma and cocoa diets on serum specific antibodies 

  1. Line: 255-261: In the opinion of the reviewer, the description of the degree of changes and significance of the results obtained in Figure 2 is not fully understandable, clear in reading for the reader - and thus gives the impression that it may not fully reflect the description of the results obtained in section 3.2. and vice versa. Yes, there is information that “Values ​​not sharing letters denote significant differences between groups (p <0.05), while values ​​sharing the same letter do not differ”. On the other hand, both the description in the legend of the figure and the marking of the bars do not reflect the degree of significance changes between the groups, both in relation to the negative control group (healthy animals), to the control asthmatic animals (not receiving any diet), and between individual groups of asthmatic rats under the influence of studied diets. Please consider whether under the graph it would be better to adopt such a system of description and notations on the graphs that would reflect the above-mentioned relationships (i.e. type <0.05; <0.005; <0.001 or something like that). 

For example, looking at the graph in Fig. 2 a - e - you can see clear differences in the bars between the CMC and control A groups (reflecting the pathological state model), not to mention that obviously relative to the REF group). The same seems to be true for APC vs A (a) or OC vs A (e). The question is whether the observed changes are statistically significant in relation to each other? What, in the eyes of the reader, does not seem fully legible from the chart, yet, is mentioned in the text.

However, this is just a suggestion to think about. The same considerations apply to the description of the results and to the legend below the figures in later sections of the manuscript.

Answer:

We agree with the reviewer that using different letters to denote significant differences could be rare for people that is not used to it. However, this way to show significant differences could be found in many articles, in figures or tables, even in the same journal (Nutrients 2022, 14, 309; Nutrients 2021, 13, 3981; Nutrients 2020, 12, 2301; Biomolecules 2021, 11, 1266; Food Chem Toxicol 2019, 127, 101; J Nutr 2018, 148, 464). The use of letters makes easy to show differences between more than two groups. For example, for Figure 2a, the group REF has an ‘a’ and the groups A, OC, APC and CMC have a ‘b’. This means that the values from group REF were statistically different from the other ones, but as ‘b’ is in the other 4 groups, there were not found significant differences between them. On the contrary, in Figure 2b, the ‘c’ letter in the top of OC, APC and CMC groups means that these values were statistically different from those of group REF (with an ‘a’ in the top) and those of group A (with a ‘b’ in the top).

When there are the same letters, no differences were found. For example, in Figure 2a there were no differences between group A and groups CMC or APC as suggested by the reviewer.

Nutrients 2022, 14(2), 309; https://doi.org/10.3390/nu14020309

Nutrients 2021, 13 (11), 3981  https://doi.org/10.3390/nu13113981

Nutrients 2020, 12(8), 2301; https://doi.org/10.3390/nu12082301

Biomolecules 2021, 11(9), 1266; https://doi.org/10.3390/biom11091266

Food Chem Toxicol 2019, 127, 101; https://pubmed.ncbi.nlm.nih.gov/30851367/

J Nutr 2018, 148, 464; https://pubmed.ncbi.nlm.nih.gov/29546302/

Discussion

  1. The authors observed an interesting relationship regarding the scale of cell infiltration in the respiratory tissues and bronchoalveolar exudates. A technical question arises at this point, have the authors considered analyzing significant, neutrophil-selective (and more of that cell fraction for which the most interesting relationships were observed) potential markers (i.e. mentioned in the manuscript: EPO, ECP)? If not, are they considering addressing this topic in future studies?. These are, of course, questions to consider and do not require a separate entry in the text of the discussion.

Answer:

It would be very interesting to know about these variables. In fact, we are planning to do it in further experiments, and this is included in the new version of the manuscript (lines 421-423).

  1. Perhaps it is worth considering including information in the discussion on the fact that the CMC diet had the most significant impact on the observed concentrations of the investigated (and detected) interleukins, which would rather be its advantage in this case.

Answer:

Following this suggestion, it has been included in the Discussion section (lines 433-435).

  1. As a comment, I would like to ask if the authors are considering carrying out similar studies taking into account the exact photochemical profile of the individual components of the cocoa paste. What in the context of the discussion, as the authors rightly emphasized at the end of the chapter, in particular with regard to flavonoids (and other potential components), seems to be intriguing? Wouldn't you consider including this information somewhere at the end of this chapter?

Answer:

Following this suggestion, this has been included in the Discussion section (lines 463-466).

  1. Isn't it also worth considering studies with a controlled dosing of the tested cocoa pastes by dosing them with an intragastric tube to animals? In this way, the researcher would have control over the application of the planned dose of each paste form?

Answer:

This approach would be very interesting and in fact we used it in our first studies (J. Agric. Food Chem. 2007, 55, 6431-6438; Clin Exp Immunol 2007, 149, 535–542; J Nutr Biochem 2008, 19, 555–565; Anal Bioanal Chem 2009, 394, 1545–1556). However, we consider that it will be very useful if we consider cocoa as a nutraceutical. Nevertheless, in our current experiments we use to give cocoa as part of diet in order to establish its beneficial effects on health (as occurs for blood pressure (EFSA J. 2014, 12, 3654).

Agric. Food Chem. 2007, 55, 6431-6438; https://pubmed.ncbi.nlm.nih.gov/17630760/

Clin Exp Immunol 2007, 149, 535–542; https://www.ncbi.nlm.nih.gov/pmc/articles/PMC2219332/

J Nutr Biochem 2008, 19, 555–565; https://pubmed.ncbi.nlm.nih.gov/18061430/

Anal Bioanal Chem 2009, 394, 1545–1556; https://pubmed.ncbi.nlm.nih.gov/19333587/

EFSA J. 2014, 12, 3654. https://doi.org/10.2903/j.efsa.2014.3654

Questions 3 and 4 are only comments for authors to consider.